# Low Serum Branched-chain Amino Acid and Insulin-Like Growth Factor-1 Levels Are Associated with Sarcopenia and Slow Gait Speed in Patients with Liver Cirrhosis

**DOI:** 10.3390/jcm9103239

**Published:** 2020-10-10

**Authors:** Chisato Saeki, Tomoya Kanai, Masanori Nakano, Tsunekazu Oikawa, Yuichi Torisu, Masayuki Saruta, Akihito Tsubota

**Affiliations:** 1Division of Gastroenterology and Hepatology, Department of Internal Medicine, The Jikei University School of Medicine, 3-25-8 Nishi-shimbashi, Minato-ku, Tokyo 105-8461, Japan; tomoyaaust@hotmail.com (T.K.); masanori-nakano@jikei.ac.jp (M.N.); oitsune@jikei.ac.jp (T.O.); torisu@jikei.ac.jp (Y.T.); m.saruta@jikei.ac.jp (M.S.); 2Division of Gastroenterology, Department of Internal Medicine, Fuji City General Hospital, 50 Takashima-cho, Fuji-shi, Shizuoka 417-8567, Japan; 3Core Research Facilities, Research Center for Medical Science, The Jikei University School of Medicine, 3-25-8 Nishi-Shimbashi, Minato-ku, Tokyo 105-8461, Japan

**Keywords:** liver cirrhosis, sarcopenia, slow gait speed, BCAA, IGF-1

## Abstract

Branched-chain amino acid (BCAA) and insulin-like growth factor 1 (IGF-1) are essential for muscle protein synthesis. We investigated the association of serum BCAA and IGF-1 levels with sarcopenia and gait speed in 192 patients with liver cirrhosis (LC). Sarcopenia was diagnosed according to the Japan Society of Hepatology criteria. Slow gait speed was defined as <1.0 m/s. Subjects were divided into three groups based on baseline BCAA or IGF-1 levels: low (L), intermediate (I), and high (H) groups. The L-BCAA group had the highest prevalence of sarcopenia (60.4%, *p* < 0.001) and slow gait speed (56.3%, *p* = 0.008), whereas the H-BCAA group had the lowest prevalence of sarcopenia (8.5%, *p* < 0.001). The L-IGF-1 group showed the highest prevalence of sarcopenia (46.9%, *p* < 0.001), whereas the H-IGF-1 group had the lowest prevalence of sarcopenia (10.0%, *p* < 0.001) and slow gait speed (18.0%, *p* = 0.003). Using the optimal BCAA and IGF-1 cutoff values for predicting sarcopenia (372 μmol/L and 48.5 ng/mL, respectively), the sensitivity and specificity were 0.709 and 0.759 for BCAA and 0.636 and 0.715 for IGF-1, respectively. Low serum BCAA and IGF-1 levels were associated with sarcopenia and slow gait speed in patients with LC.

## 1. Introduction

Sarcopenia, defined as a loss of skeletal muscle mass and strength, is associated with health-related quality of life, physical disability, and high mortality. Sarcopenia has now become the focus of attention in patients with liver cirrhosis (LC) [1,2,3]. Given that the liver plays a vital role in protein, glucose, lipid, and energy metabolism, LC is frequently complicated by protein–energy malnutrition (PEM), which can lead to sarcopenia [1]. In real-world clinical settings, the prevalence of sarcopenia in patients with LC ranges from 16.7% to 36.1% [4,5,6,7]. Accordingly, early comprehensive assessment and treatment intervention for sarcopenia are indispensable in patients with LC.

Muscle protein synthesis is positively regulated through the protein kinase B (PKB/Akt)-mediated mammalian target of the rapamycin (mTOR) signaling pathway [8,9]. Muscle growth requires the differentiation and proliferation of satellite cells, which are precursors to new muscle fibers [9]. Branched-chain amino acid (BCAA), especially leucine, is involved in muscle protein synthesis and activation of satellite cells through the PKB/Akt and mTOR pathways; therefore, it is important for maintaining and increasing muscle mass [8,9,10,11]. As PEM and hyperammonemia accelerate the consumption of BCAA by skeletal muscles for energy source and ammonia detoxification, the levels of circulating BCAA are reduced in patients with LC [12].

Insulin-like growth factor 1 (IGF-1), produced primarily in hepatocytes and some tissues including muscle, also activates the mTOR pathway via PKB/Akt and is involved in muscle protein synthesis and proliferation of satellite cells along with BCAA [9]. IGF-1 levels are decreased in patients with LC [13], resulting in reduced PKB/Akt-mediated activation of muscle growth [9].

Recently, we have reported that decreased levels of BCAA and IGF-1 were significant, independent factors associated with sarcopenia in patients with LC [4]. To further develop these findings in the present study, we investigated the association between BCAA, IGF-1, and sarcopenia in more detail and evaluated the diagnostic performance of BCAA and IGF-1 for sarcopenia in patients with LC.

## 2. Materials and Methods

### 2.1. Study Design and Patients

This was a cross-sectional study including 192 Japanese patients who were diagnosed with LC at the Jikei University School of Medicine (Tokyo, Japan) and Fuji City General Hospital (Shizuoka, Japan) between 2017 and 2020. The inclusion criteria were as follows: (1) presence of LC, (2) measurement of skeletal muscle mass index (SMI) using a bioelectrical impedance analysis (BIA) (InBody S10; InBody, Seoul, Korea), and (3) measurement of handgrip strength using a dynamometer (T.K.K5401 GRIP-D; Takei Scientific Instruments, Niigata, Japan). We excluded patients who had preexisting refractory massive ascites or implants and had been undergoing hemodialysis, as previously described [4]. This study was conducted in accordance with the Declaration of Helsinki and was approved by the Ethics Committee of the Jikei University School of Medicine (Approval no. 28–196) and Fuji City General Hospital (Approval no. 156). Written informed consent was obtained from all enrolled patients.

### 2.2. Diagnosis of Sarcopenia and Slow Gait Speed

The criteria proposed by the Japan Society of Hepatology was the basis of sarcopenia diagnosis [1]. Briefly, sarcopenia was defined as having low handgrip strength (<26 kg for men; <18 kg for women) and low muscle mass (SMI < 7.0 kg/m^2^ for men; SMI < 5.7 kg/m^2^ for women). SMI was calculated as the sum of the muscle mass of the four limbs divided by the height in square meters (kg/m^2^). Slow gait speed was a characteristic of low-physical performance and was defined as <1.0 m/s. We identified the slow gait speed by measuring gait speed over a distance of 6 m.

### 2.3. Clinical and Laboratory Assessments

Blood samples were obtained from each patient after overnight fasting. The following serum parameters were measured using routine laboratory methods: total bilirubin, albumin, and zinc levels, and prothrombin time/international normalized ratio. Serum BCAA and IGF-1 were measured using an enzymatic method (TOYOBO, Osaka, Japan) and an immunoradiometric assay (Fujirebio, Tokyo, Japan), respectively.

### 2.4. Classification Based on Serum BCAA and IGF-1 Levels

As described in the following, the median value of serum BCAA levels for all patients was 399 (interquartile range, 330–475) µmol/L. On the basis of these quartiles, patients were classified into three groups: (1) low-BCAA (L-BCAA) group, ≤330 µmol/L (first quartile); (2) intermediate-BCAA (I-BCAA) group, between 330 and 475 µmol/L (third quartile); and (3) high-BCAA (H-BCAA) group, ≥475 µmol/L (Appendix A). Similarly, as the median value of serum IGF-1 levels for all patients was 55 (interquartile range, 41–73) ng/mL, subjects were classified into three groups: (1) low-IGF-1 (L-IGF-1) group, ≤41 ng/mL (first quartile); (2) intermediate-IGF-1 (I-IGF-1) group, between 41 and 73 ng/mL (third quartile); and (3) high-IGF-1 (H-IGF-1) group, ≥73 ng/mL (Appendix A).

### 2.5. Statistical Analysis

Continuous variables are presented as medians and interquartile ranges in parentheses. The Mann–Whitney U test was used to evaluate differences in the distribution of continuous variables between two groups. The Kruskal–Wallis test, followed by the Steel–Dwass post-hoc test, was used for multiple comparisons among the three groups. Categorical variables are presented as numbers and percentages in parentheses. The chi-squared test was used to evaluate the group differences in categorical variables. The Cochran–Armitage trend test was used to assess the association between a variable with two categories and a variable with multiple categories. Additionally, univariate and multiple logistic regression analyses were performed to identify variables that were significantly and independently associated with sarcopenia. Correlations between two continuous variables were analyzed using the Spearman’s rank correlation test. To estimate the presence or absence of sarcopenia, the area under the receiver operating characteristic (ROC) curves of BCAA and IGF-1 were depicted and the optimal cutoff values were determined by the Youden index [14]. Furthermore, the sensitivity, specificity, positive predictive value (PPV), and negative predictive value (NPV) were calculated. Statistical analyses were performed using SPSS version 26 (IBM, Armonk, NY, USA), with a *p* value < 0.05 indicating statistical significance.

## 3. Results

### 3.1. Patient Characteristics

Baseline clinical characteristics of the 192 patients with LC enrolled in the present study are shown in Table 1. This study cohort included 129 males (67.2%), with a median age of 69 (59–76) years. The median IGF-1 and BCAA values were 55 (41–73) ng/mL and 399 (330–475) μmol/L, respectively. Among the 192 patients, 36 (18.8%) received oral BCAA supplementation. The median SMI and handgrip strength values were 6.95 (5.98–7.87) kg/m^2^ and 25.9 (18.3–35.3) kg, respectively. The median gait speed was 1.06 (0.86–1.26) m/s, and the prevalence of slow gait speed was 38.0% (73/192).

### 3.2. Comparison of Clinical Characteristics between Patients with and without Sarcopenia

The prevalence of sarcopenia among the 192 patients was 28.6% (55/192; Table 1). Patients with sarcopenia were older and had a lower body mass index (BMI) than those without sarcopenia (*p* < 0.001 for both). Regarding biochemical parameters, the sarcopenia group had significantly lower levels of albumin (*p* = 0.035), IGF-1 (*p* < 0.001), and BCAA (*p* < 0.001) than the non-sarcopenia group. The sarcopenia group showed a significantly higher prevalence of slow gait speed than the non-sarcopenia group (80.0% vs. 21.2%; *p* < 0.001).

### 3.3. Factors Associated with Sarcopenia in Patients with LC

The following five variables showed a significant relationship with sarcopenia in the univariate analysis: age, BMI, albumin, IGF-1, and BCAA (Appendix A). In the multivariate analysis, the following four variables were retained as independent factors associated with sarcopenia (Table 2): older age (odds ratio (OR), 1.102; 95% confidence interval (CI), 1.053–1.153; *p* < 0.001); lower BMI (OR, 0.760; 95% CI, 0.650–0.887; *p* = 0.001); lower IGF-1 (OR, 0.962; 95% CI, 0.938–0.987; *p* = 0.003); and lower BCAA (OR, 0.989; 95% CI, 0.983–0.994; *p* < 0.001).

### 3.4. Clinical Characteristics of Patients Based on Serum BCAA Levels

The prevalence of L-BCAA, I-BCAA, and H-BCAA was 25.0% (48/192), 50.5% (97/192), and 24.5% (47/192), respectively (Table 3). Among the three groups, the L-BCAA group had the highest prevalence of female patients and Child–Pugh class B/C (i.e., decompensated LC; *p* < 0.001 for both). The median levels of BMI (*p* = 0.002), albumin (*p* < 0.001), prothrombin time (*p* = 0.012), IGF-1 (*p* = 0.001), and zinc (*p* < 0.001) were significantly different among the three groups. SMI and handgrip strength values significantly decreased stepwise with a reduction in the BCAA level (Table 3, Figure 1A,B). These tendencies were also seen when stratified by gender, although they became weaker than those found in the overall cohort (probably because of a decrease in the number of patients analyzed) (Table 3, Appendix A). Notably, the L-BCAA group showed the highest prevalence of sarcopenia (60.4% (29/48); *p* < 0.001; adjusted residual = |5.6|) and slow gait speed (56.3% (27/48); *p* = 0.008; adjusted residual = |3.0|) among the three groups, whereas the H-BCAA group had the lowest prevalence of sarcopenia (8.5% (4/47); *p* < 0.001; adjusted residual = |3.5|) (Table 3, Figure 1C,D). The prevalence of sarcopenia and slow gait speed significantly increased stepwise with a reduction in the BCAA level (*p* < 0.001 and *p* = 0.004, respectively).

Next, we assessed the correlation between serum BCAA levels and baseline characteristics using the Spearman’s rank correlation test (Appendix A). Serum BCAA levels were significantly correlated with the following baseline factors: BMI, Child–Pugh score, albumin, prothrombin time, IGF-1, zinc, SMI, handgrip strength, and gait speed. The correlation coefficients for sarcopenia-related variables were highest among those for the baseline characteristics.

### 3.5. Clinical Characteristics of Patients Based on IGF-1 Levels

The prevalence of L-IGF-1, I-IGF-1, and H-IGF-1 was 25.5% (49/192), 48.4% (93/192), and 26.0% (50/192), respectively (Table 4). Among the three groups, the L-IGF-1 group had the highest prevalence of Child–Pugh class B/C (*p* = 0.005). The median levels of BMI (*p* = 0.036), albumin (*p* = 0.002), prothrombin time (*p* < 0.001), BCAA (*p* = 0.001), and zinc (*p* = 0.026) were significantly different among the three groups. The values of SMI and handgrip strength decreased stepwise with a reduction in the IGF-1 level among the overall, male, and female patients (Table 4, Figure 2A,B, Appendix A). Notably, the L-IGF-1 group showed the highest prevalence of sarcopenia (46.9% (23/49); *p* < 0.001; adjusted residual = |3.3|) among the three groups, whereas the H-IGF-1 group had the lowest prevalence of sarcopenia (10.0% (5/50); *p* < 0.001; adjusted residual = |3.4|) and slow gait speed (18.0% (9/50); *p* = 0.003; adjusted residual = |3.4|) (Table 4, Figure 2C,D). The prevalence of sarcopenia and slow gait speed significantly increased stepwise with a reduction in the IGF-1 level (*p* < 0.001 and *p* = 0.003, respectively).

Next, we assessed the correlation between serum IGF-1 levels and baseline characteristics (Appendix A). Serum IGF-1 levels were significantly correlated with the following baseline factors: Child–Pugh score, albumin, prothrombin time, BCAA, SMI, handgrip strength, and gait speed. The correlation coefficient for gait speed was highest among those for the baseline characteristics.

### 3.6. Optimal Cutoff Values of BCAA and IGF-1 for Predicting Sarcopenia

An ROC curve analysis was performed to determine the optimal cutoff values of BCAA and IGF-1, distinguishing between sarcopenia and non-sarcopenia (Figure 3). The area under the ROC (AUC) value for BCAA was 0.76. The BCAA cutoff value for predicting sarcopenia was 372 μmol/L, while the sensitivity, specificity, PPV, and NPV were 0.709, 0.759, 0.542, and 0.867, respectively. Similarly, the AUC value for IGF-1 was 0.72. The optimal IGF-1 cutoff value, its sensitivity, specificity, PPV, and NPV were 48.5 ng/mL, 0.636, 0.715, 0.473, and 0.831, respectively.

## 4. Discussion

The present study is the first to focus on the relationship between baseline serum BCAA/IGF-1 levels and sarcopenia/physical performance in patients with LC. In addition, this study has the advantages of demonstrating the stepwise changes in SMI and handgrip strength along with serum BCAA/IGF-1 levels and encompassing all three aspects of the muscle (mass, strength, and function) for investigation.

In this study, we demonstrated that the prevalence of sarcopenia among the patients with LC was 28.6% and confirmed our previous findings that lower serum levels of BCAA and IGF-1 were significantly and independently associated with sarcopenia in patients with LC [4]. Furthermore, on the basis of baseline serum BCAA or IGF-1 levels, we classified the patients into three groups and investigated the association of these levels with sarcopenia and slow gait speed. Intriguingly, SMI and handgrip strength values significantly decreased stepwise with a reduction in the BCAA level. The L-BCAA group had a significantly higher prevalence of sarcopenia and slow gait speed, whereas the H-BCAA group had a significantly lower prevalence of sarcopenia. The prevalence of sarcopenia and slow gait speed significantly increased stepwise with a reduction in the BCAA level. Similarly, SMI and handgrip strength values decreased stepwise with a reduction in the IGF-1 level. The L-IGF-1 group showed a significantly higher prevalence of sarcopenia, whereas the H-IGF-1 group had a significantly lower prevalence of sarcopenia and slow gait speed. The prevalence of sarcopenia and slow gait speed significantly increased stepwise with a reduction in the IGF-1 level. Additionally, serum BCAA and IGF-1 levels were significantly and positively correlated with SMI, handgrip strength, gait speed, and liver functional reserve including albumin and prothrombin time. These results suggest that lower serum BCAA and IGF-1 levels are closely associated with a reduction in muscle mass and strength and physical performance and could be affected by liver functional reserve in patients with LC.

Muscle mass and function are regulated by various genes and transcription factors, which are involved in protein synthesis, differentiation and proliferation of satellite cells, and proteolysis [1,9]. Notably, both BCAA (especially leucine) and IGF-1 activate the mTOR pathway via PKB/Akt and recruitment and proliferation of satellite cells, thereby promoting muscle protein synthesis and growth. Leucine-enriched BCAA supplementation has been reported to restore impaired mTOR signaling and increase autophagy, leading to reduced muscle breakdown in patients with alcoholic LC [15]. Clinical studies on community-dwelling older adults reported that the BCAA levels among subjects with sarcopenia were significantly lower than those among subjects without sarcopenia [16,17]. Other studies reported that lower IGF-1 levels were associated with sarcopenia and deteriorated physical performance in older adults [18,19,20]. Regarding proteolysis, two major ways for skeletal muscle proteolysis exist: the ubiquitin-proteasome pathway (UPP) and the autophagy system [9]. PKB/Akt signaling inhibits proteolysis via the UPP, and its inactivation and systemic inflammation activate the UPP [9]. These basic and clinical findings suggest that decreased BCAA and IGF-1 levels could contribute to the loss of muscle mass and strength and deteriorated physical performance through the unactivated mTOR pathway, suppressing satellite cell proliferation and enhancing proteolysis.

Myostatin, a cytokine belonging to the transforming growth factor-β family, is a negative regulator of satellite cell proliferation and muscle protein synthesis [1,9]. Importantly, IGF-1 signaling has dual functions in that it inhibits myostatin and activates the mTOR pathway, thereby stimulating muscle growth. Plasma myostatin levels were found to be significantly elevated in patients with LC compared with healthy controls [21]. Reportedly, higher myostatin levels were correlated with muscle mass loss and decreased BCAA to tyrosine ratio levels in patients with LC [22]. These results suggest that decreased IGF-1 and BCAA levels in patients with LC could contribute to loss of muscle mass and strength and deteriorated physical performance through the unactivated mTOR pathway and upregulated myostatin expression levels. However, the present study did not examine the myostatin levels and the association of myostatin with BCAA, IGF-1, and sarcopenia.

This study had some limitations. First, we did not assess the patients’ nutritional intake and physical exercise. Leucine-enriched nutrients with resistance exercise activate the mTOR signaling pathway and protein synthesis in skeletal muscle [23]. BCAA supplementation and walking exercise improve muscle volume and strength in patients with LC [24]. Therefore, a large-scale trial of nutritional and exercise interventions is needed to establish a treatment strategy for sarcopenia in patients with LC, especially those with low serum BCAA and IGF-1 levels. Second, patients with refractory ascites who are potentially susceptible to sarcopenia were excluded from the present study because of the unreliability of the BIA method [25]. Although SMI values estimated by the BIA method are strongly correlated with those measured using a computed tomography method with dedicated software, the former could overestimate SMI under massive ascites and thus requires careful interpretation in such cases [1]. Finally, the present study included 36 patients receiving BCAA supplementation, which could affect the BCAA levels. However, one-third of these patients were in the L-BCAA group, and there was no significant difference in the rate of BCAA supplementation among the three groups. Therefore, it could be stated that BCAA supplementation did not affect the BCAA levels in the present study (Table 1 and Table 3).

## 5. Conclusions

In the present study, we demonstrated that baseline serum BCAA and IGF-1 levels were associated with sarcopenia and slow gait speed, and that they may be surrogate markers for predicting these complications in patients with LC. Comprehensive assessment including serum BCAA and IGF-1 levels and early and effective treatment intervention are crucial in patients with LC.

## Figures and Tables

**Figure 1 jcm-09-03239-f001:**
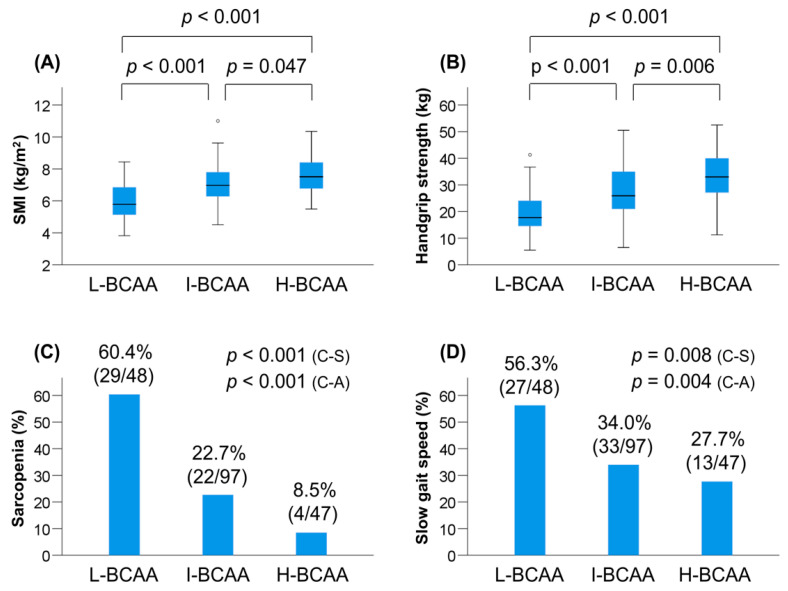
Comparison of clinical characteristics among the low (L)-branched-chain amino acid (BCAA), intermediate (I)-BCAA, and high (H)-BCAA groups. (**A**) Skeletal muscle mass index (SMI) significantly decreased stepwise with a reduction in the BCAA level. (**B**) Handgrip strength significantly decreased similarly to SMI. (**C**) The L-BCAA group significantly had the highest prevalence of sarcopenia (*p* < 0.001 by the chi-squared test), whereas the H-BCAA group significantly had the lowest prevalence of sarcopenia (*p* < 0.001 by the chi-squared test). The prevalence of sarcopenia and slow gait speed significantly increased stepwise with a reduction in the BCAA level (*p* < 0.001 by the Cochran–Armitage trend test). (**D**) The L-BCAA group showed the highest prevalence of slow gait speed (*p* = 0.008 by the chi-squared test). The prevalence of slow gait speed significantly increased stepwise with a reduction in the BCAA level (*p* = 0.004 by the Cochran–Armitage trend test). C-A, Cochran–Armitage trend test; C-S, chi-squared test.

**Figure 2 jcm-09-03239-f002:**
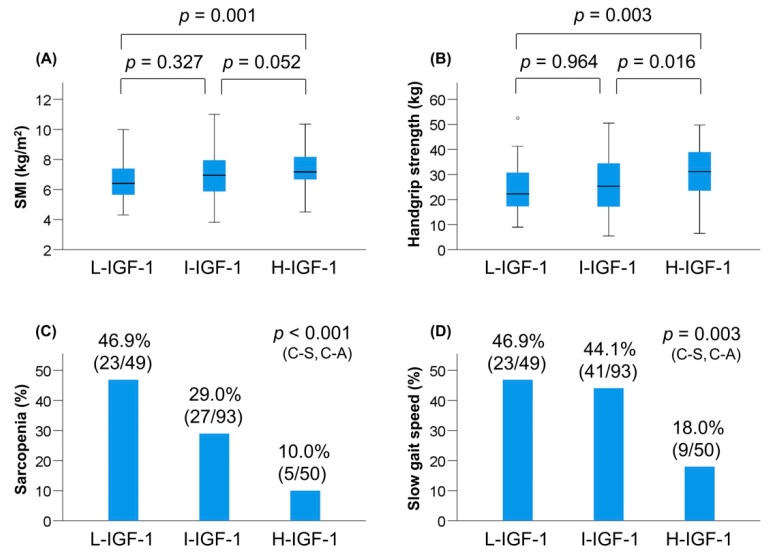
Comparison of clinical characteristics among the low (L)-insulin-like gwowth factor 1 (IGF-1), intermediate (I)-IGF-1, and high (H)-IGF-1 groups. (**A**) Skeletal muscle mass index (SMI) values were significantly lower in the L-IGF-1 group than in the H-IGF-1 group. (**B**) Handgrip strength values were significantly lower in the L-IGF-1 group than in the H-IGF-1 group. (**C**) The L-IGF-1 group significantly showed the highest prevalence of sarcopenia (*p* < 0.001 by the chi-squared test), whereas the H-IGF-1 group significantly had the lowest prevalence of sarcopenia (*p* < 0.001 by the chi-squared test). The prevalence of sarcopenia significantly increased stepwise with a reduction in the IGF-1 level (*p* < 0.001 by the Cochran–Armitage trend test). (**D**) The H-IGF-1 group significantly had the lowest prevalence of slow gait speed (*p* = 0.003 by the chi-squared test). The prevalence of slow gait speed significantly increased stepwise with a reduction in the IGF-1 level (*p* = 0.003 by the Cochran–Armitage trend test). C-A, Cochran–Armitage trend test; C-S, chi-squared test.

**Figure 3 jcm-09-03239-f003:**
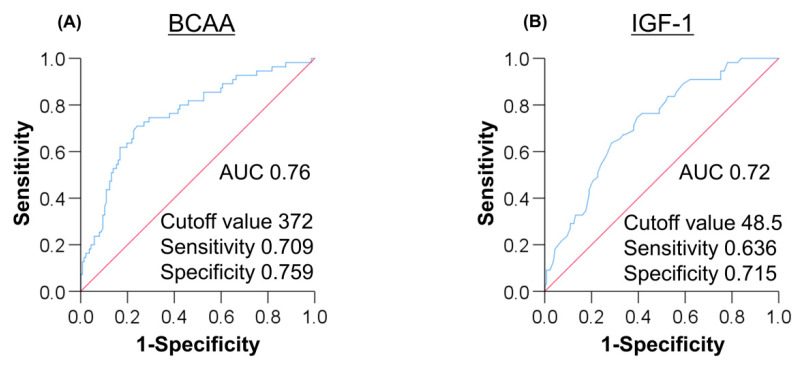
(**A**) The ROC curve analysis of branched-chain amino acid (BCAA) for predicting sarcopenia in patients with liver cirrhosis (LC). The BCAA cutoff value was 372 μmol/L with AUC, specificity, and sensitivity of 0.760, 0.709, and 0.759, respectively. (**B**) The ROC analysis of insulin-like growth factor 1 (IGF-1) for predicting sarcopenia in patients with LC. The IGF-1 cutoff value was 48.5 ng/mL with AUC, specificity, and sensitivity of 0.720, 0.636, and 0.715, respectively. AUC, area under the curve; ROC, receiver operating characteristic.

**Table 1 jcm-09-03239-t001:** Comparison of clinical characteristics between patients with and without sarcopenia.

Variable	All Patients	Sarcopenia	Non-Sarcopenia	*p* Value
Patients, n (%)	192	55 (28.6)	137 (71.4)	
Age (years)	69.0 (59.0–76.0)	76.0 (71.0–80.0)	65.0 (58.0–73.0)	<0.001
Male, n (%)	129 (67.2)	32 (58.2)	97 (70.8)	0.092
BMI (kg/m^2^)	23.7 (21.1–26.1)	21.0 (19.0–22.6)	24.5 (22.5–26.6)	<0.001
Etiology				
HBV/HCV/Alcohol/other, n	18/66/63/45	3/30/12/10	15/36/51/35	0.003
Child–Pugh A/B + C, n	129/63	35/20	94/43	0.507
Total bilirubin (mg/dL)	0.9 (0.6–1.3)	0.8 (0.5–1.4)	0.9 (0.7–1.3)	0.417
Albumin (g/dL)	3.8 (3.4–4.2)	3.7 (3.1–4.2)	3.9 (3.5–4.3)	0.035
Prothrombin time (%)	81 (67–94)	82 (68–100)	81 (67–92)	0.524
IGF-1 (ng/mL)	55 (41–73)	45 (33–55)	61 (47–77)	<0.001
BCAA (µmol/L)	399 (330–475)	321 (293–397)	425 (376–492)	<0.001
BCAA supplementation, n (%)	36 (18.8)	11 (20.0)	25 (18.2)	0.779
Zinc (µg/dL)	63 (54–74)	62 (48–73)	65 (55–74)	0.111
SMI (kg/m^2^)				
ALL patients	6.95 (5.98–7.87)	5.65 (4.99–6.44)	7.34 (6.59–8.22)	<0.001
Male	7.34 (6.79–8.25)	6.32 (5.69–6.80)	7.80 (7.14–8.46)	<0.001
Female	5.87 (5.21–6.48)	5.09 (4.55–5.39)	6.24 (5.89–6.74)	<0.001
Handgrip strength (kg)				
ALL patients	25.9 (18.3–35.3)	17.3 (14.5–23.4)	31.0 (23.3–37.2)	<0.001
Male	31.6 (25.0–37.5)	22.1 (18.3–24.2)	35.2 (30.1–40.0)	<0.001
Female	17.0 (14.2–21.9)	14.6 (12.6–16.5)	21.1 (15.3–23.4)	<0.001
Gait speed (m/s)	1.06 (0.86–1.26)	0.80 (0.61–0.97)	1.12 (1.02–1.34)	<0.001
Slow gait speed, n (%)	73 (38.0)	44 (80.0)	29 (21.2)	<0.001

Values are shown as median (interquartile range) or number (percentage). Statistical analysis was performed using the chi-squared test or the Mann–Whitney U test, as appropriate. BCAA, branched-chain amino acid; BMI, body mass index; HBV, hepatitis B virus; HCV, hepatitis C virus; IGF-1, insulin-like growth factor 1; SMI, skeletal muscle mass index.

**Table 2 jcm-09-03239-t002:** Factors associated with sarcopenia in patients with liver cirrhosis.

Variable	Univariate		Multivariate	
	OR (95%CI)	*p* Value	OR (95% CI)	*p* Value
Age (years)	1.079 (1.041–1.117)	<0.001	1.102 (1.053–1.153)	<0.001
BMI (kg/m^2^)	0.707 (0.622–0.803)	<0.001	0.760 (0.650–0.887)	0.001
Albumin (g/dL)	0.470 (0.267–0.826)	0.009		
IGF-1 (ng/mL)	0.960 (0.942–0.978)	<0.001	0.962 (0.938–0.987)	0.003
BCAA (µmol/L)	0.989 (0.985–0.993)	<0.001	0.989 (0.983–0.994)	<0.001

BCAA, branched-chain amino acid; BMI, body mass index; CI, confidence interval; IGF-1, insulin-like growth factor 1; OR, odds ratio.

**Table 3 jcm-09-03239-t003:** Characteristics of the three groups based on serum BCAA levels.

Variable	L-BCAA	I-BCAA	H-BCAA	*p* Value
Patients, n (%)	48 (25.0)	97 (50.5)	47 (24.5)	
Age (years)	71.5 (53.0–76.0)	69.0 (61.0–76.0)	68.0 (59.0–77.0)	0.938
Male, n (%)	20 (41.7)	69 (71.1)	40 (85.1)	<0.001
BMI (kg/m^2^)	22.2 (19.9–24.1)	23.7 (21.1–26.3)	24.8 (23.0–26.9)	0.002
Etiology				
HBV/HCV/Alcohol/other, n	2/17/17/12	7/33/33/24	9/16/13/9	0.262
Child–Pugh A/B + C, n	20/28	70/27	39/8	<0.001
Total bilirubin (mg/dL)	0.9 (0.6–1.8)	0.8 (0.6–1.2)	1.0 (0.7–1.3)	0.459
Albumin (g/dL)	3.5 (3.2–4.2)	3.8 (3.5–4.2)	4.1 (3.7–4.4)	<0.001
Prothrombin time (%)	74 (56–88)	80 (68–94)	86 (76–98)	0.012
IGF-1 (ng/mL)	47 (32–61)	56 (46–74)	63 (44–81)	0.001
BCAA supplementation, n (%)	12 (25.0)	18 (18.6)	6 (12.8)	0.311
Zinc (µg/dL)	58 (46–67)	62 (54–74)	70 (61–81)	<0.001
SMI (kg/m^2^)				
ALL patients	5.79 (5.11–6.88)	6.97 (6.23–7.83)	7.51 (6.70–8.43)	<0.001
Male	6.93 (6.26–7.96)	7.16 (6.82–8.21)	7.60 (7.03–8.52)	0.059
Female	5.39 (4.72–5.88)	5.99 (5.69–6.66)	6.49 (6.11–7.15)	<0.001
Handgrip strength (kg)				
ALL patients	17.8 (14.5–24.1)	25.9 (20.9–35.1)	33.0 (27.1–40.5)	<0.001
Male	24.8 (18.4–35.2)	30.7 (24.9–36.9)	35.2 (28.8–40.6)	0.004
Female	15.4 (14.1–18.0)	18.1 (14.2–23.0)	21.9 (17.9–26.0)	0.058
Sarcopenia, n (%)	29 (60.4)	22 (22.7)	4 (8.5)	<0.001
Gait speed (m/s)	0.95 (0.64–1.13)	1.09 (0.90–1.32)	1.10 (0.99–1.29)	0.001
Slow gait speed, n (%)	27 (56.3)	33 (34.0)	13 (27.7)	0.008

Values are shown as median (interquartile range) or number (percentage). Statistical analysis was performed using the chi-squared test or the Kruskal–Wallis test, as appropriate. BCAA, branched-chain amino acid; BMI, body mass index; HBV, hepatitis B virus; HCV, hepatitis C virus; IGF-1, insulin-like growth factor 1; SMI, skeletal muscle mass index.

**Table 4 jcm-09-03239-t004:** Characteristics of the three groups based on serum IGF-1 levels.

Variable	L-IGF-1	I-IGF-1	H-IGF-1	*p* Value
Patients, n (%)	49 (25.5)	93 (48.4)	50 (26.0)	
Age (years)	70.0 (57.5–77.5)	72.0 (63.0–76.0)	65.0 (56.5–73.0)	0.089
Male, n (%)	31 (63.3)	62 (66.7)	36 (72.0)	0.644
BMI (kg/m^2^)	22.5 (20.3–25.7)	23.6 (20.8–25.9)	24.8 (22.5–26.8)	0.036
Etiology				
HBV/HCV/Alcohol/other, n	3/21/18/7	6/33/27/27	9/12/18/11	0.066
Child–Pugh A/B + C, n	24/25	66/27	39/11	0.005
Total bilirubin (mg/dL)	1.1 (0.6–1.8)	0.9 (0.6–1.3)	0.8 (0.6–1.1)	0.177
Albumin (g/dL)	3.6 (3.3–3.9)	3.9 (3.4–4.3)	4.1 (3.7–4.4)	0.002
Prothrombin time (%)	66 (57–89)	81 (69–94)	88 (76–100)	<0.001
BCAA (µmol/L)	346 (311–425)	402 (333–479)	433 (390–493)	0.001
BCAA supplementation, n (%)	9 (18.4)	21 (22.6)	6 (12.0)	0.302
Zinc (µg/dL)	58 (49–70)	65 (56–73)	70 (56–78)	0.026
SMI (kg/m^2^)				
ALL patients	6.41 (5.62–7.42)	6.95 (5.86–7.98)	7.18 (6.66–8.19)	0.002
Male	7.02 (6.35–7.86)	7.38 (6.91–8.36)	7.61 (7.00–8.57)	0.049
Female	5.39 (4.87–6.08)	5.84 (5.22–6.11)	6.60 (6.06–6.96)	0.003
Handgrip strength (kg)				
ALL patients	22.2 (16.8–32.3)	25.4 (17.1–34.8)	31.2 (23.4–39.4)	0.002
Male	29.8 (21.5–35.5)	30.9 (25.2–37.0)	35.1 (29.4–42.3)	0.019
Female	15.0 (14.1–21.0)	16.5 (14.0–21.9)	22.9 (16.6–25.8)	0.022
Sarcopenia, n (%)	23 (46.9)	27 (29.0)	5 (10.0)	<0.001
Gait speed (m/s)	1.00 (0.69–1.12)	1.04 (0.80–1.21)	1.17 (1.04–1.40)	<0.001
Slow gait speed, n (%)	23 (46.9)	41 (44.1)	9 (18.0)	0.003

Values are shown as median (interquartile range) or number (percentage). Statistical analysis was performed using the chi-squared test or the Kruskal–Wallis test, as appropriate. BCAA, branched-chain amino acid; BMI, body mass index; HBV, hepatitis B virus; HCV, hepatitis C virus; IGF-1, insulin-like growth factor 1; SMI, skeletal muscle mass index.

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
