# Peer review of "Low Serum Branched-chain Amino Acid and Insulin-Like Growth Factor-1 Levels Are Associated with Sarcopenia and Slow Gait Speed in Patients with Liver Cirrhosis"

_jcm, 2020, doi:10.3390/jcm9103239_

Round 1
Reviewer 1 Report
Chisato Saeki et al investigated the association of BCAA and IGF-1 levels with sarcopenia and gait speed in patients with LC. Results underline how these two parameters might improve the management of these patients. However, I would suggest some modifications in order to make this study suitable for a publication on this journal.
Major points:
- Figure1 and figure 2 show the median values of SMI and handgrip strenght in the 3 classes (low, intermediate and high). These graphs show the differences overall. I suggest to add also graphs differentiated for male and female, as described in the tables.
- Patients (overall) have been stratified on BCAA/IGF1 levels. I suggest to perform an additional analysis, stratifying patients on BCAA/IGF1 levels in non-sarcopenic and sarcopenic subjects, separately. I acknowledge that the numerosity of sarcopenic group would decrease a lot. However, it would be interesting to understand if differences in SMI, gait speed and handgrip, are only due to higher prevalence of sarcopenia in L-BCAA/IGF1 or if a difference is already detectable in non-sarcopenic patients.
- In the discussion, the association of BCAA and IGF-1 levels with cirrhosis hallmarks such as albumine, Child-Pugh, PT should be emphasized.
Mino points:
- In the introduction section author say "therefore , it has now become the focus of attention in patients with liver cirrhosis (LC)". "Therefore" here is conclusive of something that is explained hereafter.
- In table3 there is a type-error. M-BCAA instead of I-BCAA.
Author Response
RESPONSES TO THE REVIEWER
We are most grateful for the critical comments and helpful suggestions made by reviewers, and applaud the reviewer’s careful scrutiny of our manuscript. We have responded point-by-point to the comments raised by the reviewers, as described below.
Comments and Suggestions for Authors:
Reviewer 1:
Chisato Saeki et al investigated the association of BCAA and IGF-1 levels with sarcopenia and gait speed in patients with LC. Results underline how these two parameters might improve the management of these patients. However, I would suggest some modifications in order to make this study suitable for a publication on this journal.
Major points:
(1) Figure1 and figure 2 show the median values of SMI and handgrip strenght in the 3 classes (low, intermediate and high). These graphs show the differences overall. I suggest to add also graphs differentiated for male and female, as described in the tables.
Responses: We are deeply grateful for the reviewer’s critical comments. We drew new figures (SMI and handgrip strength in the low-, intermediate-, and high-BCAA/IGF-1 groups) for men and women, respectively, based on results described in Table 3 and 4. However, it is complicated to add these new graphs to original Figure 1 and 2; therefore, we separately created the new figures as Figure S3 and S4. Even when stratified by gender, SMI and handgrip strength values decreased stepwise with a reduction in the BCAA and IGF-1 levels. However, these tendencies became weaker than those observed in the overall cohort probably due to a decrease in the number of patients analyzed.
(2) Patients (overall) have been stratified on BCAA/IGF1 levels. I suggest to perform an additional analysis, stratifying patients on BCAA/IGF1 levels in non-sarcopenic and sarcopenic subjects, separately. I acknowledge that the numerosity of sarcopenic group would decrease a lot. However, it would be interesting to understand if differences in SMI, gait speed and handgrip, are only due to higher prevalence of sarcopenia in L-BCAA/IGF1 or if a difference is already detectable in non-sarcopenic patients.
Responses: As suggested by the reviewer, we performed an additional analysis, stratifying patients according to the BCAA/IGF1 levels in those with and without sarcopenia, respectively (Figure RX and Table RX only for review). Among patients with sarcopenia, SMI and handgrip strength values were significantly lower in the L-BCAA group than in the I-BCAA group, whereas gait speed values were not significantly different among the three BCAA groups [Table R1, Fig. R1(A)–(C)]. In addition, SMI, handgrip strength, and gait speed values were not significantly different among the three IGF-1 groups [Table R2, Fig. R1(D)–(E)]. The number of patients in each group became too small to statistically define the relationship between groups. Meanwhile, among patients without sarcopenia, handgrip strength values were significantly lower in the I-BCAA group than in the H-BCAA group, whereas SMI and gait speed values were not significantly different among the three BCAA groups [Table R3, Fig. R2(A)–(C)]. In addition, gait speed values were significantly lower in the L- and I-IGF-1 groups than in the H-IGF-1 group, whereas SMI and handgrip strength values were not significantly different among the three IGF-1 groups [Table R4, Fig. R2(D)–(E)]. Similar to results in patients with sarcopenia, the number of patients in each group became too small to statistically confirm the association between groups. These results suggest that differences in SMI, handgrip strength, and gait speed in overall patients may be due to high prevalence of L-BCAA/L-IGF-1 and very low prevalence of H-BCAA/H-IGF-1 among patients with sarcopenia. Alternatively, the findings that approximately half of patients without sarcopenia had intermediate BCAA/IGF-1 levels suggest that they may have low BCAA/IGF-1 levels and sarcopenia in the near future.
(3) In the discussion, the association of BCAA and IGF-1 levels with cirrhosis hallmarks such as albumine, Child-Pugh, PT should be emphasized.
Responses: We thank the reviewer for the above suggestion. We emphasized the association of BCAA and IGF-1 levels with liver functional reserve, including albumin and prothrombin time on lines 283–288 in the revised text.
Mino points:
(1) In the introduction section author say "therefore , it has now become the focus of attention in patients with liver cirrhosis (LC)". "Therefore" here is conclusive of something that is explained hereafter.
Responses: As suggested by the reviewer, we deleted the term "therefore" from the introduction section.
(2) In table3 there is a type-error. M-BCAA instead of I-BCAA.
Responses: As pointed by the reviewer, we corrected the typographic error.
Reviewer 2 Report
I think this paper is a good study showing that BCAA levels and IGF-1 levels are dose-dependently related to sarcopenia. My major concern is as below; #1. In this study, BIA was used as a diagnostic criterion for sarcopenia. BIA can measure the amount of muscle relatively accurately in the general population, but its effectiveness has not yet been accurately verified in cases involving cirrhosis, especially ascites. Since only "refractory ascites" are included in the study exclusion criteria, it is advisable to supplement more references on the accuracy of BIA in patients with cirrhosis. #2. Are there any specific reasons for excluding "refractory ascites" patients? #3. Was this study registed such as UMIN or clinicaltrialgov? #4. There have been many hypotheses that BCAA is involved in the pathophyisology of sarcopenia in cirrhotic patients. #5. The advantages of this study are that 1) the dose-dependency of BCAA and IGF-1 was demonstrated, and 2) all three aspects of muscle were presented (muscle mass, muscle strength, functional sarcopenia). It would be better to emphasize this point a little more. #6. There are 37 patients taking oral BCAA preparations, and it seems that it may have affected the measurement of BCAA levels. It would be better to describe more in limitation.
Author Response
RESPONSES TO THE REVIEWER
We are most grateful for the critical comments and helpful suggestions made by reviewers, and applaud the reviewer’s careful scrutiny of our manuscript. We have responded point-by-point to the comments raised by the reviewers, as described below.
Comments and Suggestions for Authors:
Reviewer 2:
I think this paper is a good study showing that BCAA levels and IGF-1 levels are dose-dependently related to sarcopenia. My major concern is as below;
(1) In this study, BIA was used as a diagnostic criterion for sarcopenia. BIA can measure the amount of muscle relatively accurately in the general population, but its effectiveness has not yet been accurately verified in cases involving cirrhosis, especially ascites. Since only "refractory ascites" are included in the study exclusion criteria, it is advisable to supplement more references on the accuracy of BIA in patients with cirrhosis.
Responses: We are deeply grateful for the reviewer’s critical comments. A previous report demonstrated that SMI values estimated using the BIA method were strongly correlated with those measured using a CT method with a dedicated software in patients with chronic liver disease, including those with LC complicated by ascites (Ref. 1). However, the BIA method could overestimate SMI under the massive ascites. We newly described the application and relevance of the BIA method for muscle mass assessment in patients with LC on lines 325–328 in the revised text.
(2) Are there any specific reasons for excluding "refractory ascites" patients?
Responses: We are deeply grateful for the reviewer’s comments. We excluded patients who had refractory massive ascites, because the BIA analysis could overestimate SMI in such patients. There are no other reasons for excluding patients with refractory massive ascites.
(3) Was this study registed such as UMIN or clinicaltrialgov?
Responses: We are deeply grateful for the reviewer’s comments. This study was approved by the Ethics Committee of the Jikei University School of Medicine and Fuji City General Hospital. However, we did not register the present study with UMIN or clinicaltrialgov, because it was not an interventional study.
(4) There have been many hypotheses that BCAA is involved in the pathophyisology of sarcopenia in cirrhotic patients.
Responses: We are deeply grateful for the reviewer’s critical comments. Both impaired protein synthesis and proteolysis in skeletal muscle cause the loss of muscle mass and strength, leading to sarcopenia. BCAA (especially leucine) and IGF-1 activate the mTOR pathway via PKB/Akt and promote muscle protein synthesis. Two major skeletal muscle proteolytic pathways are the ubiquitin-proteasome pathway (UPP) and the autophagy system. PKB/Akt also inhibits proteolysis via UPP. Furthermore, a previous report has demonstrated that BCAA supplementation improved the impaired mTOR signaling and increased autophagy, leading to reduced muscle breakdown in patients with alcoholic LC (Tsien C, et al. Hepatology, 2015; new Ref. 20). Taken together, decreased BCAA and IGF-1 levels could contribute to the loss of muscle mass and strength and deteriorated physical performance through unactivated mTOR pathway and promoted proteolysis. We newly added these descriptions (the involvement of BCAA in the pathophysiology of sarcopenia) to the revised discussion section (lines 293–305).
(5) The advantages of this study are that 1) the dose-dependency of BCAA and IGF-1 was demonstrated, and 2) all three aspects of muscle were presented (muscle mass, muscle strength, functional sarcopenia). It would be better to emphasize this point a little more.
Responses: We are deeply grateful for the reviewer’s kindly suggestions. We emphasized these points suggested by the reviewer and newly added the descriptions to the revised discussion section (lines 264–268).
(6) There are 37 patients taking oral BCAA preparations, and it seems that it may have affected the measurement of BCAA levels. It would be better to describe more in limitation.
Responses: We are deeply grateful for the reviewer’s critical comments. The present study included 36 patients receiving BCAA supplementation, which could affect the BCAA levels. However, BCAA supplementation has not affected the distribution of groups stratified by the BCAA levels (Table 3) and the prevalence of sarcopenia in the present study (Table 1). We newly added this limitation on lines 328–332 in the revised text.
Round 2
Reviewer 2 Report
All of the points pointed out have been well corrected.
Author Response
Comments and Suggestions for Authors
Reviewer2:
(1) All of the points pointed out have been well corrected.
Responses: We are deeply grateful for the reviewer’s encouraging comments.